# National Trends and Disparities in Complementary Food Diversity Among Infants: A 12-Year Cross-Sectional Birth Cohort Study

**DOI:** 10.3390/nu17040636

**Published:** 2025-02-11

**Authors:** Eun Lee, Seonkyeong Rhie, Ju Hee Kim, Eun Kyo Ha, Min Seo Kim, Won Suk Lee, Boeun Han, Man Yong Han

**Affiliations:** 1Department of Pediatrics, Chonnam National University Hospital, Chonnam National University Medical School, Gwangju 61469, Republic of Korea; unelee@daum.net; 2Department of Pediatrics, CHA Bundang Medical Center, CHA University School of Medicine, Seongnam 13496, Republic of Korea; starclusters@gmail.com (S.R.); h.boeunn@gmail.com (B.H.); 3Department of Pediatrics, Kyung Hee University Medical Center, Kyung Hee University College of Medicine, Seoul 02447, Republic of Korea; 2004052@gmail.com; 4Department of Pediatrics, Hallym University Kangnam Sacred Heart Hospital, Seoul 07441, Republic of Korea; dmsry1@gmail.com; 5School of Medicine, Gyeongsang National University, Jinju 52727, Republic of Korea; mikoesnim91@gmail.com; 6Department of Pediatrics, CHA Ilsan Medical Center, CHA University, Goyang 10414, Republic of Korea; leeerped@gmail.com

**Keywords:** infant, complementary food, diversity, disparity, atopic dermatitis, food allergy

## Abstract

**Background:** The complementary food introduction and consumption guidelines for atopic dermatitis and food allergy prevention have evolved; however, their impact on infant feeding practices remains unclear. This study aimed to analyze complementary food diversity trends in infants, identify vulnerable infants with limited food diversity, and examine the trends in infants with or without vulnerable factors over time. **Methods:** This study analyzed infants aged 9–12 months who participated in the food diversity survey, conducted as part of the National Health Screening Program in Korea from 2009 to 2020. The complementary food items included grains, vegetables, fruits, eggs, fish, and meats. Infants consuming “six” and “less than six” complementary food items were categorized into high- and low-food-diversity groups, respectively. The study employed logistic regression models to examine the trends in food diversity and vulnerable factors with an assessment of the interaction effects. **Results:** This study included 3,425,301 participants (51.5% male) aged 11.3 months (standard deviation, 0.8). The high-food-diversity prevalence significantly increased over time, from 30.8% in 2009 to 52.9% in 2020 (*p* < 0.001). Vulnerable infants included those with preterm birth, low birth weight, non-breastfeeding status, high socioeconomic status, non-Seoul residence at birth, any perinatal conditions, hospitalization due to wheezing, atopic dermatitis and food allergies. The high-diversity proportion increased significantly over the study period across all vulnerable factors (*p* for interaction < 0.001). However, no significant interactions were observed between the study years and vulnerable factors, except for food allergy (*β* Coefficient, −0.0117, *p* for interaction = 0.004). **Conclusions:** The increasing trends in high-complementary-food-diversity proportions highlight the substantial progress over the study period. However, persistent disparities in vulnerable populations underline the importance of targeted interventions, including tailored nutritional education and policies, that promote equitable dietary practices during early life.

## 1. Introduction

Complementary food diversity is essential for the nutritional and developmental outcomes of infants, significantly impacting their growth, immune function, and long-term health [1]. Food diversity during infancy can be shaped by various factors, including cultural practices, socioeconomic status, and parental perceptions. Vegetables and fruits supply essential vitamins, minerals, and fiber, contributing to immune health and digestion [2]. Protein-rich foods such as meats, fish, and eggs are critical for muscle development, iron supplementation, and the prevention of micronutrient deficiencies [2,3]. In particular, fish provides omega-3 fatty acids, which are essential for brain development, while eggs serve as a key source of high-quality protein and choline [4]. Although complementary foods provide essential nutrients, certain food groups, such as egg whites, were previously considered highly allergenic and were recommended for delayed introduction [5].

In 2000, the American Academy of Pediatrics (AAP) issued a key guideline recommending the delay of the introduction of allergenic foods (e.g., eggs) until after 1 year of age to mitigate the risk of allergic disease [5]. This recommendation has long influenced feeding practices, despite subsequent evidence challenging its efficacy in allergy prevention [6]. Introducing diverse complementary foods during infancy not only supports optimal growth and development but also serves a critical role in immune system maturation [1]. Consequently, the AAP later revised its guidelines, advising against food restrictions and advocating for the early introduction of assorted foods [6,7].

Despite the updated guidelines on complementary food practice and education, caregivers of infants with vulnerable factors, such as the presence of allergic diseases and premature birth, often encounter heightened anxiety regarding the introduction of multiple complementary foods, leading to delayed or restricted dietary diversity [8,9]. The hesitation, compounded by limited access to reliable information or the fear of potential adverse reactions, reportedly contributes to disparities in food diversity between infants with and without vulnerable factors [10]. In addition, socioeconomic, geographic, and clinical factors may further affect complementary food introduction patterns in infancy [10,11]. However, research on these patterns during infancy is lacking, and comprehensive data on the real-world implementation of these guideline revisions are scarce.

Therefore, we aimed to investigate the temporal consumption trends of six major complementary foods, namely, vegetables, meats, grains, fruits, fish, and eggs, in terms of food diversity in infants aged 9–12 months between 2009 and 2020, following the transition in dietary guidelines. In addition, we sought to elucidate the determinants of complementary food diversity in early life. Determining the consumption trends, along with their associated factors, of diverse complementary foods can aid in improving dietary practices and formulating targeted intervention strategies that promote healthy and diversified dietary practice patterns in early life.

## 2. Methods

### 2.1. Study Population

This study was performed as part of the National Investigation of Birth Cohort in Korea 2008 (NICKs-2008), a nationwide, population-based health screening program initiated in 2008 [12]. A total of 3,518,398 infants born between 2008 and 2021 who had undergone regular check-ups at 9–12 months of age were initially considered for the study. Owing to the relatively small number of participants who had completed surveys on complementary food introduction patterns during infancy in 2008 (n = 18,804) and 2021 (n = 74,293), participants born in 2008 and 2021 were excluded from the analysis. Resultantly, 3,425,301 infants born in 2009–2020 who had participated in the surveys on complementary food introduction patterns at 9–12 months were included in the final analysis. Dietary information was collected through caregiver- and parent-reported questionnaires during clinic visits for regular check-ups at 9–12 months of age. This study utilized de-identified individual data authorized for research purposes under the National Health Insurance Act. Ethical approval for the research was obtained in accordance with the Act and relevant guidelines and regulations. The study protocol was reviewed and approved by the Institutional Review Board of the Korea National Institute for Bioethics Policy (P01-201603-21-005). As the study used anonymized, publicly available data, patient consent was not mandatory. The Institutional Review Board of the Korea National Institute for Bioethics Policy waived the requirement for informed consent.

### 2.2. Dietary Practices

The dietary practices during the complementary feeding introduction period were obtained at 9–12 months of age using the following questionnaire item: “Which types of food do you provide as complementary foods to your baby? (1) grains, (2) vegetables, (3) fruits, (4) eggs, (5) fish, and/or (6) meats”. Caregivers or parents could select all that apply upon completing the questionnaire. The participants were categorized into high- (introduction of six complementary food items) and low- (introduction of less than six complementary food items) food-diversity cohorts based on the number of introduced complementary food items at check-up.

### 2.3. Definitions

Atopic dermatitis (AD) was defined as having five or more principal diagnoses based on the International Classification of Diseases, 10th Revision, (ICD-10) code for AD (L20.9), reflecting its chronic, relapsing, and remitting inflammatory nature to ensure a more accurate identification of AD cases [13]. Food allergy (FA) was identified based on a principal diagnosis with any of the following ICD-10 codes: T78.0, T78.1, L23.6, L24.6, L25.4, L27.2, and/or K52.294. A wheezing episode was defined as hospital admission with a principal diagnosis based on any of the following ICD-10 codes: J45.X, J46.X, and/or J21.X. All three conditions were restricted to participants diagnosed before the administration of the solid food survey.

The preterm status, defined as birth before 37 weeks of pregnancy, and birth weight were determined based on caregiver-reported questionnaire responses. The breastfeeding status was assessed using survey items inquiring about feeding practices within the first 4 months of life, with participants categorized into the “exclusive breastfeeding” group if they had received breastmilk only. Perinatal conditions were identified using ICD-10 codes ranging from P10 to P96, which include congenital malformations and chromosomal anomalies. Socioeconomic status was determined using insurance income quintiles and categorized into three groups: low (≤25th percentile), intermediate (25th–75th percentile), and high (>75th percentile). Birth residence was classified as Seoul, metropolitan, city, or rural, following the regional classification system of South Korea.

### 2.4. Statistical Analysis

The demographic characteristics and clinical variables of the study population are summarized as frequencies and percentages for categorical variables and means with standard deviations for continuous variables. To more effectively elucidate the prevalence trend of each demographic variable from 2009 to 2020, the study period was divided into three intervals: 2009–2012, 2013–2016, and 2017–2020.

Reversed stacked area plots were utilized to visually present the cumulative fraction of individuals across complementary food diversity categories. The changes in prevalence rates over time were analyzed by dividing the population into two groups: those who consumed all six types of complementary foods at 9–12 months and those who consumed less than six. To evaluate the trends in the consumption of each complementary food item over time, a logistic regression analysis was performed. The year (2009–2020) was used as an independent variable, whereas the consumption of each complementary food item (grains, vegetables, fruit, eggs, fish, and meats) was treated as a binary dependent variable. The adjusted odds ratios (aORs) and their 95% confidence intervals (CIs) were calculated to assess the direction and magnitude of each trend. All models were adjusted for potential covariates, including sex (male or female), birth weight (continuous), breastfeeding during the first 4 months (yes or no), prematurity (yes or no), AD or FA (yes or no), wheezing episodes (yes or no), birth residence (Seoul or rural), socioeconomic status (tertile: low, intermediate, or high), and certain conditions originating in the perinatal period (yes or no).

The factors associated with low food diversity in early life were analyzed using logistic regression models. AORs with 95% CIs were calculated for each variable after adjusting for confounding factors, such as sex, birth weight, prematurity, breastfeeding during the first 4 months, birth residence, household income level, and any perinatal conditions. The interaction terms (e.g., year *×* vulnerable factor) were included in the regression models to ascertain whether the relationships between vulnerable factors and high food diversity altered over time. The statistical significance was set at *p* < 0.05, and all analyses were conducted using SAS (version 9.4; SAS Institute, Inc., Cary, NC, USA).

## 3. Results

### 3.1. Baseline Characteristics of the Study Population

The study population comprised 3,425,301 infants whose demographic characteristics are summarized in Table 1. Their median age at the time of the survey was 12 months. The sex distribution was consistent across time periods: 51.5% males and 48.5% females. Birth weight slightly decreased over time (3.2 kg in 2009–2012, 3.18 kg in 2013–2016, and 3.17 kg in 2017–2020). The prevalence of prematurity increased over time from 5.61% (2009–2012) to 7.16% (2017–2020). The breastfeeding rate in the first 4 months exhibited a notable decrease (32.5% in 2009–2012, 30.2% in 2013–2016, and 19.2% in 2017–2020). Most infants were born in cities (48.3%), followed by metropolitan areas (24.5%), Seoul (19.7%), and rural areas (6.8%). Socioeconomic status (income quintiles) remained stable across the three periods. Conditions originating in the perinatal period increased over time. The prevalence of AD displayed a decreasing trend from 24.7% (2009–2012) to 11.0% (2017–2020). The prevalence of hospitalization owing to wheezing or FA diagnosis remained stable over time.

### 3.2. Trends in Complementary Food Diversity in Infancy from 2009 to 2020

Figure 1 illustrates the longitudinal trends in the number of introduced complementary food items in the total population, demonstrating a significantly increasing trend of infants exposed to a greater number of complementary food items (*p* < 0.0001). The proportion of infants introduced to all six complementary food items markedly increased from 30.8% in 2009 to 52.9% in 2020 (*p* for trend < 0.001). However, the proportion of infants introduced to three or four types of complementary foods significantly decreased, while no statistically significant changes were observed for those introduced to two or five types of complementary foods over the same period (Figure 1 and Appendix A).

### 3.3. Consumption Trend of Each Complementary Food Item by Year

There was a steady increase in the consumption rates of grains, fruits, eggs, fish, and meats, with fish and fruits showing the most pronounced increase (Figure 2). Conversely, vegetable consumption exhibited a slight decline, despite being consistently high. Table 2 presents the aORs and 95% CIs for the consumption of each complementary food item by year. The odds of consuming fish (aOR = 1.0374; 95% CI: 1.0369–1.0379; *p* < 0.0001), fruits (aOR = 1.0345; 95% CI: 1.0341–1.0349; *p* < 0.0001), eggs (aOR = 1.0207; 95% CI: 1.0202–1.0212; *p* < 0.0001), meats (aOR = 1.0019; 95% CI: 1.0015–1.0023; *p* < 0.0001), and grains (aOR = 1.0034; 95% CI: 1.003–1.0038; *p* < 0.0001) increased significantly over time in infants during the study period. However, the odds of vegetable (aOR = 0.9973; 95% CI: 0.9969–0.9977; *p* < 0.0001) consumption decreased significantly over time, although vegetables exhibited the highest consumption rate among the six complementary food items (Figure 2 and Appendix A).

### 3.4. Factors Associated with the Introduction of Low Complementary Food Diversity in Infancy

Compared with infants exhibiting high complementary food diversity (introduction of all six complementary food items), those displaying low food diversity (five or fewer items) were associated with several demographic, socioeconomic, and clinical factors (Table 3). Infants with a birth weight below 2.5 kg (aOR = 1.018, 95% CI: 1.009–1.027) and preterm birth (aOR = 1.009, 95% CI: 1.001–1.017) were more likely to have low food diversity. Socioeconomic factors also played a role, with infants from high-income families (aOR = 1.027, 95% CI: 1.018–1.063) and those born in metropolitan regions (aOR = 1.088, 95% CI: 1.080–1.097), cities (aOR = 1.054, 95% CI: 1.044–1.063), and rural areas (aOR = 1.045, 95% CI: 1.034–1.057) having higher odds of low food diversity compared with those born in Seoul. Clinically, infants with FA (aOR = 1.156, 95% CI: 1.138–1.174), wheezing (aOR = 1.029, 95% CI: 1.025–1.032), or AD (aOR = 1.085, 95% CI: 1.080–1.089) were significantly associated with low food diversity. Breastfeeding for <4 months was also a contributing factor (aOR = 1.041, 95% CI: 1.038–1.045). Food diversity demonstrated an improving trend over time, as indicated by a significant decrease in the odds of low food diversity in later years (aOR = 0.9554, 95% CI: 0.9549–0.9559).

### 3.5. Interaction Effects Between Birth Year and Complementary Food Diversity Across Vulnerable Factors

The high diversity proportion increased significantly over the study period across all vulnerable factors (*p* for interaction < 0.001). After identifying the vulnerable factors associated with low food diversity (Table 3), the interaction effects between “year” and “each vulnerable factor” were analyzed to examine the trends in high complementary food diversity over time (Table 4). An analysis of the interaction effects between birth year and high complementary food diversity revealed no significant interactions for most of the vulnerable factors, including sex, birth weight, prematurity, breastfeeding in the first 4 months, birth residence, socioeconomic status, wheezing, and AD. However, a significant negative interaction was observed for FA (β = −0.0117, *p* = 0.004).

## 4. Discussion

This study provides comprehensive data on complementary food diversity trends from 2009 to 2020 in infants aged 9–12 months, highlighting the overall improvements while identifying persistent disparities across sociodemographic and health-related factors. The rate of improvement in complementary food consumption in infancy varied across food items; the consumption of grains, fruits, eggs, fish, and meats exhibited increasing trends over the study period, with substantial increases in fish and fruit intakes. Infants with preterm birth, low birth weight, lack of breastfeeding, high socioeconomic status, residence outside Seoul at birth, perinatal conditions, hospitalization for wheezing, atopic dermatitis, and food allergy were associated with low complementary food diversity in infancy. This study highlights the steady increase in high complementary food diversity rates across most vulnerable groups over the study period with slower improvements in dietary diversity in infants with FA. The findings emphasize the importance of tailored interventions for vulnerable populations, particularly those with food allergies, to ensure equitable improvements in dietary diversity.

A comprehensive investigation of the consumption trends of each complementary food item provides insight into how specific food items have been incorporated into infant diets. Over the study period, the consumption of grains, fruits, eggs, fish, and meats significantly increased. Notably, the substantial increase in fish and egg consumption during infancy suggests a shift toward dietary diversity in infancy that aligns with public health guidelines promoting healthy nutrition for infants. This trend potentially reflects enhanced nutrition education, greater food accessibility, and evolving parental practices [14]. Vegetables exhibited the highest consumption rate during the study period, although the odds of vegetable consumption declined slightly over the same period. Although the absolute vegetable intake levels during infancy may remain comparatively high, barriers to vegetable consumption, such as taste preference, accessibility, and socioeconomic factors, need to continue being addressed. Continued efforts are required to sustain high complementary food diversity in infancy, thus ensuring balanced nutrition during early life.

The overall increase in complementary food diversity reflects positive progress in nutritional practices over time, as evidenced by a significant rise in the introduction of diverse food items, such as fish, fruits, and eggs. In particular, the increase in complementary food diversity in infants with AD or FA over time is attributable to evolving nutritional guidelines and their practical implementation. However, despite this progress, infants with FA demonstrated a significantly slower improvement in the prevalence of high complementary food diversity over the study period. In addition, infants with FA exhibited the lower prevalence of high complementary food diversity among all groups, displaying a persistent disparity over time. Caregivers of these infants may avoid introducing complementary foods owing to the risk of exacerbating symptoms or triggering allergic reactions, despite emerging evidence corroborating early and diverse food introduction to prevent allergic disease development [6,7,15]. In addition, infants with wheezing before check-up at 9–12 months exhibited a significantly lower prevalence of high complementary food diversity compared with those without wheezing, and this disparity demonstrated minimal improvement over time. Wheezing episodes are often associated with respiratory infections or early signs of asthma, potentially leading to conservative feeding practices among caregivers, including the delayed introduction of complementary foods owing to fears of triggering allergic or adverse reactions. The persistent inequality suggests a need for improved education on desirable complementary dietary practices for infants with allergic diseases or those at high risk of allergic disease. Collaborative efforts among pediatricians, allergists, and nutritionists can empower caregivers to navigate dietary challenges and promote diverse feeding practices, even in the context of allergic disease.

While the overall trend in complementary food diversity reflects progress, the lack of significant interaction effects between study years and most of the vulnerable factors suggests that improvements were experienced uniformly rather than disproportionately benefiting specific high-risk groups. However, the observed disparity in food diversity trends for infants with FA underscores the unique challenges faced by this group. Despite evolving nutritional guidelines encouraging early allergen introduction, caregivers of FA infants may remain cautious, leading to slower improvements in dietary diversity. The significant negative interaction effect observed for FA supports this concern, indicating that the gap in food diversity between FA and non-FA infants persisted over time. This suggests that general improvements in nutritional practices may not be equally effective for all subpopulations, necessitating more targeted public health efforts. Beyond FA, the absence of significant interaction effects for other vulnerable factors suggests that while food diversity improved, existing disparities in feeding practices were not necessarily reduced. Future research should investigate whether specific barriers, such as misinformation, access to allergen-safe dietary options, or healthcare guidance, contribute to these trends and explore strategies to enhance the effectiveness of dietary interventions for at-risk infants.

Infants with perinatal conditions demonstrated consistently low complementary food diversity compared with those without complications, with the disproportion slightly widening over time. This trend suggests that infants with perinatal issues may encounter unique challenges, such as delayed feeding initiation, medical complications, and prolonged hospital stays [16], potentially impacting early complementary dietary practices. Enhanced medical care and caregiver education over the study period may explain the improvement in complementary food diversity in infancy. Nevertheless, efforts to improve complementary food diversity in infants with perinatal conditions are still warranted.

Infant diet quality with respect to complementary food diversity is important in infant growth and development [17]. A previous cross-sectional study with a smaller sample size reported a lack of dietary diversity in preterm infants during the second half of infancy, especially in those of a low socioeconomic status [18]. In the present study, preterm infants demonstrated a consistently low prevalence of high complementary food diversity compared with full-term infants throughout the study period. Despite the overall improvements in complementary food diversity for both groups, preterm infants lagged behind, underlining ongoing disparities in dietary practices. Prematurity may also be associated with feeding difficulties in early life, developmental delays, and increased parental anxiety regarding complementary food introduction, which may collectively hinder the achievement of high complementary food diversity. The findings of the present study underscore the burgeoning need for targeted nutritional education and support for caregivers of preterm infants, especially considering the increasing prevalence of preterm births in recent years [19]. Socioeconomic factors, including income level and residence area at birth, can significantly influence complementary food diversity in infants owing to a complex interplay between socioeconomic status and complementary feeding practices. Therefore, comprehensive strategies that address economic, educational, and cultural factors to improve complementary dietary diversity are requisite to enhancing nutritional practice in infancy [20]. In this study, a high socioeconomic status was defined as having a household income in the upper 25th percentile. However, income alone may not fully capture dietary practices, as factors such as caregiver education, cultural influences, and food accessibility also play critical roles [20,21]. Future research should consider integrating these dimensions to provide a more comprehensive understanding of the relationship between socioeconomic status and complementary food diversity in infants.

Our study discovered that infants who had been breastfed during the first 4 months exhibited high complementary food diversity. Another study found that infants breastfed for >6 months tend to be introduced to complementary foods later, although their complementary food diversity does not significantly differ from that of formula-fed infants [22]. In contrast, another study reported lower dietary diversity among infants who were predominantly breastfed [23]. These studies suggest that complementary food diversity in breastfed infants is influenced by a combination of factors beyond breastfeeding itself, including regional variations, socioeconomic status, and the impact of nutritional education. These findings underscore the potential role of tailored nutritional strategies and education programs in improving complementary feeding practices, rather than solely attributing outcomes to breastfeeding duration.

Notwithstanding, this study has certain limitations. This study lacks detailed information on dietary patterns, including the introduction timing of each complementary food and frequency of complementary food consumption, limiting our ability to assess their impact on complementary food diversity trends. In addition, parental dietary preferences and family disease history, both of which can significantly influence complementary food introduction patterns, were not accounted for in our analysis. The reliance on self-reported questionnaires introduces short-term recall bias, which may affect the reporting of allergic food introduction and frequency of consumption. The results of the present study have limitations in the interpretation of causal relationships between complementary food introduction patterns and health-related conditions, such as wheezing, AD, or FA. This cross-sectional study provides valuable insights into national trends but does not allow for the assessment of individual changes in complementary feeding practices over time. A prospective cohort study would be needed to examine the evolution of early feeding behaviors and their impact on health outcomes, including allergic diseases.

External influences, including cultural shifts, economic fluctuations, and public health initiatives, were not explicitly accounted for in our analysis. The increasing trend in complementary food diversity may have been influenced by government nutrition programs, evolving infant feeding recommendations, changes in food marketing, and the COVID-19 pandemic, which altered household food availability and caregiver behaviors. Future research should integrate these external determinants into models assessing dietary trends. While this study provides a comprehensive analysis of complementary food diversity trends in South Korea, its findings may not be fully generalizable to other countries with different dietary habits, cultural attitudes, and socioeconomic structures. Furthermore, marginalized populations, including those with lower healthcare access, may be underrepresented in the National Health Screening Program dataset. Future research should prioritize data collection from underrepresented groups to ensure equitable policy recommendations. To address these limitations and reduce disparities in complementary feeding practices, targeted interventions are necessary. Educating caregivers on the benefits and safety of introducing diverse foods early, training healthcare professionals to offer evidence-based nutritional guidance, and implementing food accessibility programs for disadvantaged families could help improve dietary diversity across all socioeconomic groups.

## 5. Conclusions

Our findings indicate a significant improvement in complementary food diversity among Korean infants from 2009 to 2020. However, disparities persist, particularly among infants with FA and other vulnerable groups. To address these gaps, targeted interventions should include caregiver education on the benefits and safety of diverse complementary food introduction, improved training for healthcare providers to support caregivers in making informed feeding choices, and policies that improve food accessibility for socioeconomically disadvantaged families. Additionally, future research should incorporate external sociocultural and economic influences into dietary diversity trends and employ longitudinal designs to better understand the long-term impact of complementary feeding practices on health outcomes. These measures can promote equitable nutrition and better health outcomes for all infants, particularly those in vulnerable populations.

## Figures and Tables

**Figure 1 nutrients-17-00636-f001:**
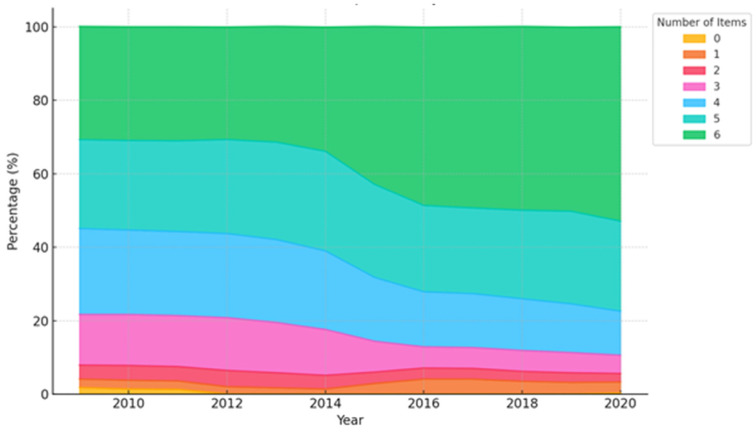
Trends in the number of consumed complementary food items between 2009 and 2020 in the total population.

**Figure 2 nutrients-17-00636-f002:**
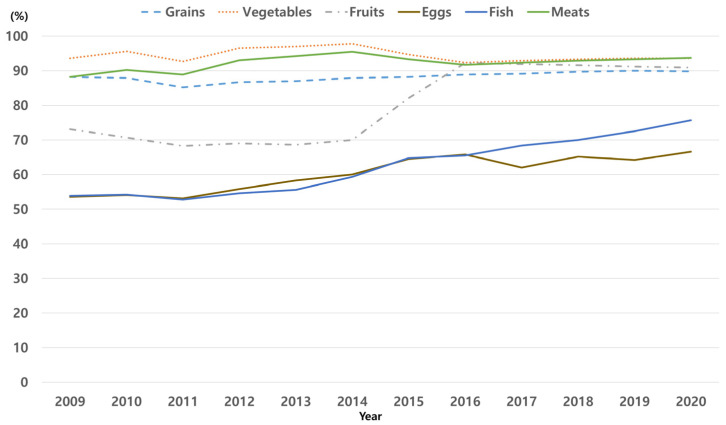
Trends in the consumption rates of complementary food items among infants aged 9–12 months during the study period.

**Table 1 nutrients-17-00636-t001:** Demographic characteristics of the study population.

n (%) or Mean (SD)	Total		2009–2012		2013–2016		2017–2020	
Total number	3,425,301	100.0%	975,997	28.5%	1,299,823	37.9%	1,152,481	33.6%
Male	1,764,310	51.5%	502,486	51.5%	668,782	51.5%	593,042	51.5%
Female	1,660,991	48.5%	470,511	48.2%	631,041	48.5%	559,439	48.5%
Age at check-up, median (Q1–Q3) months	12.0	(11–12)	11.0	(10–12)	12.0	(11–12)	12.0	(11–13)
Birth weight, mean (SD), kg	3.18	(0.45)	3.2	(0.44)	3.18	(0.44)	3.17	(0.46)
Body weight upon first examination, mean (SD), kg	8.12	(1.62)	8.13	(1.52)	8.09	(1.48)	8.14	(1.83)
Body weight upon second examination, mean (SD), kg	9.88	(1.67)	9.83	(1.48)	9.84	(1.36)	9.98	(2.1)
Prematurity	204,309	5.96%	54,787	5.61%	67,037	5.16%	82,485	7.16%
Breastfeeding in the first 4 months	930,724	27.17%	317,166	32.50%	392,549	30.20%	221,009	19.18%
Residence at birth								
Seoul	674,397	19.7%	211,772	21.7%	251,113	19.3%	211,512	18.4%
Metropolitan	840,215	24.5%	232,252	23.8%	324,745	25.0%	283,218	24.6%
City	1,655,396	48.3%	450,748	46.2%	627,122	48.2%	577,526	50.1%
Rural	233,355	6.8%	67,000	6.9%	86,801	6.7%	79,554	6.9%
Socioeconomic status								
Low (≤25 percentile)		24.9%	244,771	25.1%	325,360	25.0%	289,056	25.1%
Intermediate (>25 percentile and ≤75 percentile)	1,687,741	49.3%	472,516	48.4%	631,953	48.6%	568,202	49.3%
High (>75 percentile)	864,195	25.2%	244,485	25.0%	329,468	25.3%	294,552	25.6%
Condition(s) originating in the perinatal period	1,874,450	54.7%	498,802	51.1%	701,641	54.0%	671,007	58.2%
Birth trauma (P10–P159)	47,235	1.4%	10,307	1.1%	16,894	1.3%	20,034	1.7%
Respiratory and cardiovascular disorder specific to the perinatal period (P20–P299)	310,062	9.1%	67,569	6.9%	111,780	8.6%	130,713	11.3%
Infections specific to the perinatal period (P35–P399)	486,150	14.2%	140,095	14.4%	178,577	13.7%	167,478	14.5%
Hemorrhagic and hematological disorders of fetus and newborn (P50–P619)	1,255,963	36.7%	331,304	33.9%	476,984	36.7%	447,675	38.8%
Transitory endocrine and metabolic disorders (P70–P749)	238,494	7.0%	45,673	4.7%	93,120	7.2%	99,701	8.7%
Digestive system disorders of fetus and newborn (P75–P789)	67,085	2.0%	26,136	2.7%	24,600	1.9%	16,349	1.4%
Conditions involving the integument and temperature regulation (P80–P849)	138,876	4.1%	39,379	4.0%	49,735	3.8%	49,762	4.3%
Congenital malformations, deformations, and other disorders originating in the perinatal period (P90–P969)	207,166	6.0%	60,596	6.2%	75,128	5.8%	71,452	6.2%
Chromosomal anomaly	22,111	0.6%	4385	0.4%	6841	0.5%	10,885	0.9%
Hospitalization owing to wheezing before check-up	604,764	17.7%	90,730	9.3%	142,113	10.9%	129,021	11.2%
Atopic dermatitis before check-up	587,043	17.1%	241,520	24.7%	218,498	16.8%	127,025	11.0%
Food allergy before check-up	29,249	0.9%	7965	0.8%	10,846	0.8%	10,438	0.9%

SD—standard deviation.

**Table 2 nutrients-17-00636-t002:** Adjusted odds ratio and 95% confidence interval for the consumption of each complementary food item over time.

Food Item	aOR ^1^	95% Lower	95% Upper	*p* Value
Grains	1.0034	1.0030	1.0038	<0.0001
Vegetables	0.9973	0.9969	0.9977	<0.0001
Fruits	1.0345	1.0341	1.0349	<0.0001
Eggs	1.0207	1.0202	1.0212	<0.0001
Fish	1.0374	1.0369	1.0379	<0.0001
Meats	1.0019	1.0015	1.0023	<0.0001

^1^ Adjusted for confounding factors, including sex, birth weight, prematurity, breastfeeding in the first 4 months, birth residence, household income level, and any perinatal condition. aOR—adjusted odds ratio.

**Table 3 nutrients-17-00636-t003:** Factors associated with low food diversity in infancy.

Variables, n (%) or Mean (SD)	High Food Diversity ^1^		Low Food Diversity ^2^		aOR ^3^	95% Lower	95% Upper
	n	%	n	%			
Sex							
Male	716,513	40.6%	1,047,797	59.4%	Ref		
Female	672,083	40.5%	988,908	59.5%	1.005	1.002	1.008
Month at second check-up, mean (SD)	6.37	(3.39)	6.44	(3.35)	1.002	1.0015	1.0024
Birth weight, mean (SD), kg							
≥2.5 kg	1,321,650	40.6%	1,932,066	59.4%	Ref		
<2.5 kg	66,946	39.0%	104,639	61.0%	1.018	1.009	1.027
Body weight at first check-up, mean (SD), kg	8.14	(1.60)	8.11	(1.64)			
Body weight at second check-up, mean (SD), kg	9.96	(1.72)	9.83	(1.64)			
Prematurity							
No	1,307,522	40.6%	1,913,470	59.4%	Ref		
Yes	81,074	39.7%	123,235	60.3%	1.009	1.001	1.017
Breastfeeding in the first 4 months							
Yes	385,326	41.4%	545,398	58.6%	Ref		
No	808,473	42.7%	1,086,711	57.3%	1.041	1.038	1.045
Birth residence							
Seoul	287,553	42.6%	386,844	57.4%	Ref		
Metropolitan	330,195	39.3%	510,020	60.7%	1.088	1.080	1.097
City	672,576	40.6%	982,820	59.4%	1.054	1.044	1.063
Rural	91,201	39.1%	142,154	60.9%	1.045	1.034	1.057
Socioeconomic factors							
Low (≤25 P)	355,438	41.7%	495,989	58.3%	Ref		
Intermediate (>25 P and ≤75 P)	690,299	40.9%	997,442	59.1%	0.983	0.976	0.991
High (>75 percentile)	335,788	38.9%	528,407	61.1%	1.027	1.018	1.063
Year of examination, mean (SD)	2015.39	3.19	2014.29	3.22	0.9554	0.9549	0.9559
Condition(s) originating in the perinatal period							
No	626,409	40.4%	924,442	59.6%	Ref		
Yes	762,187	40.7%	1,112,263	59.3%	1.0121	1.0089	1.0153
Hospitalization owing to wheezing before check-up							
No	993,240	41.6%	1,394,788	58.4%	Ref		
Yes	395,356	38.1%	641,917	61.9%	1.029	1.025	1.032
Atopic dermatitis before check-up							
No	1,187,243	41.8%	1,651,015	58.2%	Ref		
Yes	201,353	34.3%	385,690	65.7%	1.085	1.080	1.089
Food allergy before check-up							
No	1,378,930	40.6%	2,017,122	59.4%	Ref		
Yes	9666	33.0%	19,583	67.0%	1.156	1.138	1.174

^1^ High complementary food diversity refers to the introduction of all six complementary food items. ^2^ Low complementary food diversity refers to the introduction of five or fewer complementary food items. ^3^ Adjustments were made for sex, birth weight, prematurity, breastfeeding in the first 4 months, birth residence, household income level, and any perinatal condition. aOR—adjusted odd ratio; P—percentile; Ref—reference; SD—standard deviation.

**Table 4 nutrients-17-00636-t004:** Interaction effects between birth year and high complementary food diversity rate depending on vulnerable factors.

Variables	%	2009	2010	2011	2012	2013	2014	2015	2016	2017	2018	2019	2020	*p* for Trend	*β* Coefficient	*p* for Interaction
Sex	male	30.8	30.9	32.5	30.5	31.5	34.0	43.0	48.6	47.8	50.2	50.3	53.4	<0.0001	0.0003	0.942
	female (ref)	30.8	30.9	31.0	30.6	31.4	33.7	43.1	48.5	47.6	49.9	49.8	52.4			
Birth weight	<2.5 kg	29.2	29.6	28.4	28.5	30.0	32.2	40.6	45.7	44.6	47.2	46.7	50.3	<0.0001	−0.002	0.605
	≥2.5 kg (ref)	29.8	31.0	31.1	30.7	31.5	33.9	43.2	48.7	47.9	50.2	50.3	53.1			
Prematurity	yes	30.9	31.0	31.1	30.7	31.5	33.9	43.1	48.7	48.3	50.2	50.3	53.1	<0.0001	−0.0012	0.757
	no (ref)	29.6	29.5	29.5	28.8	30.3	32.8	41.0	46.1	45.4	47.9	47.7	50.8			
BMF in the first 4 months	no	30.0	30.5	31.1	30.4	31.5	33.8	43.2	49.0	48.0	50.8	50.6	53.1	<0.0001	0.0004	0.920
	yes (ref)	33.3	32.9	32.9	33.4	33.8	36.8	46.2	51.3	51.1	52.9	53.1	56.1			
Birth residence	rural	30.4	30.5	30.3	30.0	30.8	33.3	42.3	47.8	47.1	49.4	49.4	52.1	<0.0001	0.0004	0.928
	Seoul (ref)	31.8	32.9	34.0	33.2	34.2	36.0	46.0	51.7	50.4	52.6	53.0	56.4			
Socioeconomic status	Low (ref)	31.5	32.1	33.0	32.3	33.0	35.0	44.9	50.4	49.3	51.6	52.2	55.4	<0.0001	0.0014	0.724
	high	29.3	29.1	28.7	28.3	29.7	32.3	40.9	46.4	46.0	48.5	48.8	51.7			
Any perinatal conditions	yes	30.6	30.4	30.6	30.2	31.1	33.5	42.7	48.2	47.3	49.7	49.7	52.8	<0.0001	0.0003	0.951
	no (ref)	31.1	31.5	31.5	30.9	31.9	34.2	43.5	48.9	48.2	50.5	50.5	53.0			
Wheezing	yes	30.9	30.9	30.3	29.0	29.8	32.7	41.2	46.2	46.3	48.2	48.7	51.4	<0.0001	−0.0015	0.719
	no (ref)	30.8	30.9	31.1	30.8	31.7	34.0	43.3	48.8	47.9	50.3	50.2	53.0			
AD	yes	27.7	28	28.7	28.3	28.9	31.5	39.2	43.7	42.6	45.3	44.5	46.5	<0.0001	−0.0018	0.644
	no (ref)	32.1	31.9	31.8	31.2	32.1	34.3	43.8	49.3	48.4	50.7	50.7	50.2			
FA	yes	29.8	26.2	28.1	26.3	26.7	30	35.9	37.6	37.4	36.3	37.5	39.7	<0.0001	−0.0117	0.004
	no (ref)	30.8	31	31	30.6	31.5	33.9	43.1	48.6	47.8	50.2	50.2	53.0			

AD—atopic dermatitis; BMF—breastmilk feeding; FA—food allergy; ref—reference group.

## Data Availability

This study utilized data from the National Health Claims Database, maintained by the National Health Insurance Service (NHIS) of the Republic of Korea. The access to this database was granted following a review by the NHIS Research Support Inquiry Committee. Upon approval, the raw data were made available to researchers for a fee. Owing to the intellectual property rights held by the NHIS, we were unable to share the data, analytical methods, or research materials with other researchers. However, the database is available for research purposes, and interested investigators can access it to replicate our findings by following the outlined application process (https://nhiss.nhis.or.kr/, accessed on 12 June 2024).

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
