# Peer review of "National Trends and Disparities in Complementary Food Diversity Among Infants: A 12-Year Cross-Sectional Birth Cohort Study"

_nutrients, 2025, doi:10.3390/nu17040636_

Round 1

Reviewer 1 Report

Comments and Suggestions for Authors

my comments:

1.       In my opinion, patients from 2008 and 2021 should not have been excluded from this study, the groups were quite large and such practice could have influenced the final conclusions

2.       Line 92-93: "Which types of food do you provide as complementary foods to your baby? 1) grains, 2) vegetables, 3) fruits, 4) eggs, 5) fish, and/or 6) meats." - was that the entire questionnaire? If not, please include the complete questionnaire in English and in the original language in the supplementary materials

3.       line 166 "Figure 1" - missing figure - fill in

4.       similarly line 173 "Figure 1 and Supplementary Table 1)" - missing indicated elements, no supplement

5.       line 182 - Supplementary Table 2 - no table

6.       please diversify the description of the results with graphs

7.       references - please expand the list of references, there are currently too many self-citations in relation to publications by other authors; besides, the list is very short (there are more studies available that are consistent with this topic and should be used in the discussion of the results)

Author Response

Thank you for taking the time to review our manuscript. We sincerely appreciate your valuable comments and suggestions.

We have attached our detailed responses to your comments in the accompanying file. Your feedback has greatly contributed to improving the quality of our manuscript, and we are grateful for your insightful review.

Thank you again for your time and consideration.

Reviewer 2 Report

Comments and Suggestions for Authors

The presented study presents insights into the trends of complementary food diversity among infants in South Korea from 2009 to 2020. 

While the sample size of over 3.4 million participants lends robustness to the findings, the reliance on data from a national health screening program may lead to biases. The screening may not capture all infants, particularly those from marginalized communities, potentially skewing the results.

The identification of vulnerable infants is crucial, yet the definition and parameters used to classify these groups could be further elaborated. For instance, the criteria for “high socioeconomic status” and its implications on food diversity could be more clearly defined, as wealth does not always correlate with healthy dietary practice.

The reported increase in high-food-diversity proportions is promising, but the lack of significant interactions between study years and most vulnerable factors suggests that improvements may not be uniformly experienced. This raises questions about the effectiveness of interventions and whether they truly reach those who need them most.

While the conclusions emphasize the need for targeted interventions, there is little detail on what these interventions might entail. The study would benefit from specific recommendations based on the identified gaps in dietary practices among vulnerable populations.

The manuscript's cross-sectional nature at different time points may overlook individual changes over time in complementary food practices. A longitudinal approach could provide deeper insights into how dietary habits evolve in response to interventions.

The paper does not account for external factors such as cultural shifts, economic changes, or the impact of the COVID-19 pandemic, which might have influenced dietary practices during the latter part of the study period.

Although the findings are specific to South Korea, they may not be applicable to other contexts with different cultural and socioeconomic backgrounds. Caution should be exercised when generalizing these results to other populations.

Author Response

(The authors gave the same response as above.)

Reviewer 3 Report

Comments and Suggestions for Authors

National trends and disparities in complementary food diversity among infants: A 12-year cross-sectional birth cohort study

 The study aimed to analyze complementary food diversity trends in infants, identify vulnerable infants with limited food diversity, and examine the trends in infants with or without vulnerable factors over time. This study analyzed infants aged 9–12 months who participated in the food diversity survey, conducted as part of Korea's National Health Screening Program from 2009 to 2020. The high-food-diversity prevalence significantly increased with time, from 30.8% in 2009 to 52.9% in 2020 (p < 0.001). The authors concluded that the increasing trends in high complementary food diversity proportions highlight substantial progress over the study period.

A comprehensive study, well-planned and executed. There are a few points that need to be addressed.

In the abstract: list the six major complementary foods.

In the introduction: more details are required about the advantages and disadvantages of those groups of food for infants.

L100: Atopic dermatitis (AD) was defined as having five or more principal diagnoses, give more details.

Excellent statistical analysis

Excellent data presentation.

L166: fig 1, I don't see it in the manuscript.

Author Response

(The authors gave the same response as above.)

Round 2

Reviewer 1 Report

Comments and Suggestions for Authors

I thank the Authors for the corrections they made. I have no other comments.

Author Response

We sincerely appreciate the time and effort you have dedicated to reviewing our manuscript. We are pleased to hear that our revisions have addressed your concerns, and we deeply appreciate your constructive input throughout the review process.

Once again, thank you for your time and consideration. We truly appreciate your support in enhancing the quality of our work.

Reviewer 2 Report

Comments and Suggestions for Authors

After some improvements in the manuscript, I believe it can be published.

Author Response

We sincerely appreciate your thoughtful review and constructive feedback on our manuscript. In response to your comments, we have made additional revisions to the limitations and conclusion sections to further enhance the clarity and completeness of our study.

lines 366-411: 

Notwithstanding, this study has certain limitations. This study lacks detailed information on dietary patterns, including the introduction timing of each complementary food and frequency of complementary food consumption, limiting our ability to assess their impact on complementary food diversity trends. In addition, parental dietary preferences and family disease history, both of which can significantly influence complementary food introduction patterns, were not accounted for in our analysis. The reliance on self-reported questionnaires introduces short-term recall bias, which may affect the reporting of allergic food introduction and frequency of consumption. The results of the present study have limitations in the interpretation of causal relationships between complementary food introduction patterns and health-related conditions, such as wheezing, AD, or FA. This cross-sectional study provides valuable insights into national trends, but does not allow for the assessment of individual changes in complementary feeding practices over time. A prospective cohort study would be needed to examine the evolution of early feeding behaviors and their impact on health outcomes, including allergic diseases.

External influences, including cultural shifts, economic fluctuations, and public health initiatives, were not explicitly accounted for in our analysis. The increasing trend in complementary food diversity may have been influenced by government nutrition programs, evolving infant feeding recommendations, changes in food marketing, and the COVID-19 pandemic, which altered household food availability and caregiver behaviors. Future research should integrate these external determinants into models assessing dietary trends. While this study provides a comprehensive analysis of complementary food diversity trends in South Korea, its findings may not be fully generalizable to other countries with different dietary habits, cultural attitudes, and socioeconomic structures. Furthermore, marginalized populations, including those with lower healthcare access, may be underrepresented in the National Health Screening Program dataset. Future research should prioritize data collection from underrepresented groups to ensure equitable policy recommendations. To address these limitations and reduce disparities in complementary feeding practices, targeted interventions are necessary. Educating caregivers on the benefits and safety of introducing diverse foods early, training healthcare professionals to offer evidence-based nutritional guidance, and implementing food accessibility programs for disadvantaged families could help improve dietary diversity across all socioeconomic groups.

  1. Conclusions

Our findings indicate a significant improvement in complementary food diversity among Korean infants from 2009 to 2020. However, disparities persist, particularly among infants with FA and other vulnerable groups. To address these gaps, targeted interventions should include caregiver education on the benefits and safety of diverse complementary food introduction, improved training for healthcare providers to support caregivers in making informed feeding choices, and policies that improve food accessibility for socioeconomically disadvantaged families. Additionally, future research should incorporate external sociocultural and economic influences into dietary diversity trends and employ longitudinal designs to better understand the long-term impact of complementary feeding practices on health outcomes. These measures can promote equitable nutrition and better health outcomes for all infants, particularly those in vulnerable populations.

Your insights have been invaluable in strengthening the quality of our manuscript, and we truly appreciate the time and effort you have dedicated to reviewing our work. Your feedback has played a crucial role in refining our discussion and ensuring that our findings are presented with greater precision and depth.

Thank you once again for your support in improving the quality of our manuscript.